# Third Trimester Fetuses Demonstrate Priming, a Form of Implicit Memory, In Utero

**DOI:** 10.3390/children9111670

**Published:** 2022-10-31

**Authors:** Hanna Gustafsson, Jennifer Hammond, Julie Spicer, Sierra Kuzava, Elizabeth Werner, Marisa Spann, Rachel Marsh, Tianshu Feng, Seonjoo Lee, Catherine Monk

**Affiliations:** 1Department of Psychiatry, Oregon Health and Science University, Portland, OR 97239, USA; 2Division of Neonatology, Department of Pediatrics, Columbia University Irving Medical Center, New York, NY 10032, USA; 3Department of Psychiatry, Icahn School of Medicine at Mount Sinai, New York, NY 10029, USA; 4Department of Psychology, Stony Brook University, Stony Brook, NY 11794, USA; 5Department of Psychiatry, Columbia University Irving Medical Center, New York, NY 10032, USA; 6New York State Psychiatric Institute, New York, NY 10032, USA; 7Department of Biostatistics (in Psychiatry), Mailman School of Public Health, Columbia University Irving Medical Center, New York, NY 10032, USA; 8Department of Obstetrics and Gynecology, Columbia University Irving Medical Center, New York, NY 10032, USA

**Keywords:** fetal development, memory, priming

## Abstract

Research examinations of changes in fetal heart rate (HR) to operationalize fetal memory suggests that human memory capacities emerge in utero. However, there is little evidence for a form of implicit memory or priming. The present aim was to determine if priming is evident in utero. Fetal HR, maternal HR and maternal respiratory rate (RR) were examined in 105 women during the third trimester of pregnancy. Women experienced two counterbalanced laboratory tasks, the Stroop task and the paced breathing task, and their cardiorespiratory activity functioned as a stimulus for fetuses. Repeated measures ANOVAs revealed maternal HR increased during the Stroop task but only when the Stroop task was presented first (89.64 bpm to 92.39 bpm) (*p* = 0.04). Maternal RR increased during the Stroop task, regardless of task order (17.72 bpm to 21.11 bpm; 18.50 bpm to 22.60 bpm) (*p* < 0.01). Fetal HR increased during the paced breathing task, but only when it followed maternal exposure to the Stroop task (141.13 bpm to 143.97 bpm) (*p* < 0.01). Fetuses registered maternal HR and RR reactivity to the Stroop task, which influenced their response during maternal engagement with a related task, suggesting priming. Further study of fetal memory may suggest another pathway by which prenatal exposures impact future development.

## 1. Introduction

The establishment of memory represents a critical component of infant brain development [1]. A growing body of literature suggests that memory capacities emerge in utero [2]. Implicit memory, defined as the unconscious activation of previous experiences to influence behavior, likely develops first and includes habituation, classical conditioning, and priming. Explicit memory develops later and involves the conscious awareness of previous experiences [3,4,5]. Research has utilized fetal heart rate (HR) to operationalize memory in utero [6]. Habituation, defined as a decrease in attention or response after repeated exposure, is established in fetuses by 30 weeks’ gestation [1,7]. Specifically, third-trimester fetuses appear to register, recognize, and respond to varied stimuli measured by changes in HR during habituation paradigms [7,8,9,10,11], though see DeCasper and Moon [12] for challenges to the interpretation that some of these results demonstrate fetal memory. Additional work suggests that fetuses demonstrate auditory recognition memory for maternal voice measured using fetal HR. For example, third-trimester fetuses respond with an increase in HR during recordings of their mothers’ voices and with a decrease in HR during recordings of a female stranger’s voice [9]. Previous work has also used fetal movements to demonstrate classical conditioning in utero, which occurs when an unconditioned stimulus and a neutral stimulus result in the same response after repeated pairings of the stimuli have occurred. During the third trimester, fetuses demonstrated classical conditioning by increased fetal movement after repeated exposure to a paired vibration and loud noise [1,13,14].

While fetal HR and fetal movements have been established as markers of memory in utero, few studies using these measures to investigate a third form of implicit memory, priming, exist. Priming is evident when exposure to one stimulus results in a response to a second stimulus that does not occur in the absence of exposure to the first. Research has demonstrated a variety of priming effects in infants, toddlers, and school-age children [15,16,17,18]. For example, infants at eighteen months of age exhibited sophisticated priming capabilities. Specifically, infants who were primed with imagery suggesting group affiliation demonstrated increased pro-social behavior on a subsequent task when compared to infants unexposed to the imagery [19]. In a separate study, toddlers at age twenty-one months looked longer at a target picture when the prime picture was semantically related [20]. Since priming as a form of implicit memory is crucial for attention and learning, it is important to further investigate its development in utero.

The aim of the present study was to identify evidence of priming in utero using aspects of a paradigm that we and others have developed over the past decade [21,22,23,24]. Women in the third trimester of pregnancy engage in a set order of two standard laboratory tasks, the Stroop color-word matching task and the paced breathing task, while maternal HR, respiratory rate, and fetal HR are monitored. These laboratory tasks leverage the principle of cardiorespiratory coupling, which occurs under typical physiological conditions. Cardiorespiratory coupling describes the effects of changes in respiration on both HR and blood pressure [25]. Specifically, any increase in respiratory rate leads to an increase in HR [26,27,28]. It is also established that changes in maternal respiratory rate and HR impact the fetus, since fetuses react to alterations in their in utero environment [29]. For example, increases in maternal HR in the setting of exercise lead to increases in fetal HR [30]. Both the Stroop task and the paced breathing task are associated with maternal autonomic nervous system (ANS) reactivity [31,32,33]. Our group as well as others have found that, on average, fetuses do not have any HR change during the Stroop task [21,22,24]. However, on average, fetuses respond with an increase in HR during the paced breathing task [23]. These results are based on a uniform ordering of the tasks. In these studies, the Stroop task was presented first followed by the paced breathing task, and conceptualized as a maternal task-based cardiorespiratory activity that perturbs the in utero environment and elicits a change in fetal HR.

To assess fetal priming in this study, we counterbalanced the task order. We hypothesized that three distinct patterns in the fetus could emerge. First, the paced breathing task may have unique qualities that elicit an increase in fetal HR. If this were in evidence, we would expect to see an increase in fetal HR only during the paced breathing task regardless of the task order. Second, there may be a cumulative effect of consecutive maternal ANS reactivity tasks such that fetal HR increases in response to any second task. This would be supported if fetal HR increases only during the second task, regardless of the task type. Third, only one order of the tasks would lead to an increase in fetal HR during the second task, indicating a priming effect from the first task.

## 2. Methods

### 2.1. Participants

Participants were part of a longitudinal study examining maternal and fetal epigenetic modifications in the context of maternal prenatal stress (*N* = 152). The study was approved by the institutional review board and written consent was obtained. Pregnant women ages 18–45 were recruited through the Department of Obstetrics and Gynecology at Columbia University Irving Medical Center. All women had a healthy singleton pregnancy at recruitment. Participants were excluded if they acknowledged smoking tobacco or the use of recreational drugs, or lacked fluency in English. Participants also were excluded if they used the following: nitrates, steroids, beta blockers, triptans, and psychiatric medications. Recruitment procedures allowed for enrollment in the first or second trimester.

The study design included three prenatal sessions; data used in the current analyses came from the third session during the third trimester when women were 34–37 weeks pregnant. Of the 152 women enrolled, 131 completed this session: nine (5.92%) were on bedrest or delivered prior to the session and 12 (7.89%) missed the session or withdrew from the study prior to it. An additional three participants (1.97%) completed the session but their fetal data was of poor quality. Two participants (1.32%) did not have fetal data collected due to equipment failure. The first 21 participants who enrolled in the study only completed the Stroop task. These observations were not included in the analyses, though sensitivity analyses show that our findings remain unchanged if these participants are included. The final sample included 105 participants. This subsample did not differ from the complete (*N =* 152) sample on any variables considered in the current study.

### 2.2. Procedures

During the third session, women reported demographic information and completed the fetal and maternal monitoring. As described in detail in previous publications [23,34], this process sees women recline in a semi-recumbent position while maternal and fetal physiological signals including HR are monitored, and maternal ratings of each task’s stress impact are measured. The session is divided into six periods: a 20 min resting baseline, a 5 min resting baseline, a 5 min Stroop task, a 6 min paced breathing task, and two separate 5 min recovery periods (interspersed between the two tasks). The current study used the latter five periods, consistent with prior studies [21,22,23] (Figure 1 and Figure 2).

During the Stroop task, women were presented with color words in either congruently- or incongruently-colored letters and were asked to identify the color of the letters. A computer-generated voice intoned a congruent or incongruent color and the experimenter prompted participants to work faster. The paced breathing task requires women to control their breath by inhaling when a bar on a computer screen rose and exhaling when it fell. During the task, participants alternate breathing at a rate faster than typical (e.g., 30 bpm), slower than typical (e.g., 10 bpm), and at an approximately typical rate for a pregnant woman (e.g., 20 bpm). There was no voice prompt. Participants were randomly assigned to undertake the Stroop (*N* = 51) or paced breathing task (*N* = 54) first, forming two groups based on task presentation order.

### 2.3. Measures

Demographic and Psychosocial Variables. Women reported on the following information: age at enrollment, ethnicity, race, highest level of completed education (in years), and the family’s annual income. Women reported their pre-pregnancy weight in pounds and a research assistant measured each participant’s height. Using this information, standard calculation procedures were made to produce maternal pre-pregnancy body mass index [35]. The child’s sex, gestational age (GA; in weeks), and weight (in grams) at birth were obtained from medical records. GA at the fetal session (in weeks) was calculated by comparing the date of fetal monitoring with the participant’s last menstrual period.

Fetal HR Monitoring. Fetal HR was acquired using a Toitu MT 325 fetal actocardiograph (Toitu Co., Ltd., Tokyo, Japan). The Toitu detects fetal HR via a single Doppler transducer placed on a woman’s abdomen and processes this signal through a series of filters. Fetal HR data were collected from the output port of the Toitu MT 325 fetal actocardiograph and were digitized at 50 Hz using a 16-bit A/D card (National Instruments 16XE50). Data were analyzed offline using the custom MATLAB 8.3 scripts (The Mathworks Inc., Natick, MA, USA) developed for our projects. A fetal HR below 80 or above 200 beats per minute (bpm) was linearly interpolated and then low-pass filtered at three Hz using a 16-point finite impulse response filter. The mean of the resulting fetal HR was taken over non-interpolated values. Filtered fetal HR was further examined for artifacts in the following way: times at which the absolute sample-to-sample (20 ms) change in fetal HR exceeded five bpm were found, and fetal HR was marked as an artifact until it returned to within five bpm of the previous value. The resultant gaps were linearly interpolated.

Maternal HR and Respiration Monitoring. Electrocardiogram electrodes (ECG) were placed beneath the women’s left and right clavicles, with the ground located on the lower right back. Analog ECG and respiration signals were digitized at 500 Hz and 50 Hz, respectively, by a 16-bit A/D conversion board (National Instruments, Austin, TX, USA) and passed to a microcomputer. The ECG waveform was submitted to an R-wave detection routine implemented by custom-written software, resulting in an RR interval series. Ectopic beats were corrected by interpolation. Mean HR was computed for each period in units of bpm. For the respiration waveform, peaks and troughs were identified by custom-written software and visually inspected for accuracy. Average breaths per minute were calculated for each period.

Maternal Stress Ratings. Women were asked to report on a 10-point Likert-type scale (where 1 = none at all and 10 = extreme stress) how stressed they felt at the end of each of the key periods: the resting baseline, the Stroop task, and the paced breathing task [21,22].

### 2.4. Analytic Strategy

Independent samples *t* tests and chi-squared tests were used to examine whether the two groups of participants (i.e., Stroop task first versus paced breathing task first) differed on a number of variables that previous research has associated with fetal neurobehavioral development: maternal age, ethnicity, race, education, family income, pre-pregnancy BMI, child sex, GA and weight at birth, and GA at the fetal session. Changes in fetal HR, maternal HR, maternal respiratory rate, and maternal stress ratings were tested using a series of repeated measures ANOVAs where task and group membership (Stroop task first versus paced breathing task first) were considered as predictors of fetal and maternal physiology. GA at the fetal session also was included as a covariate in the fetal HR model, consistent with previous research and in line with knowledge that fetal HR changes across pregnancy [34]. ANOVA analyses also compared fetal HR changes between groups within each period. Associations between maternal physiological responses and fetal HR were assessed using bivariate correlations. SPSS 22 was used for all analyses.

## 3. Results

### 3.1. Descriptive Information

Descriptive information about the study population is presented in Table 1. Women presented with the Stroop task first versus the paced breathing task first did not differ on any variables.

### 3.2. Group Differences in Fetal HR Reactivity

Fetal HR values during all periods are provided in Table 2. First, group comparisons within each period were performed. There were no group differences in fetal HR between the fetuses whose mothers completed the paced breathing task first or Stroop task first during the first task period (*p* = 0.59) or during the first recovery period (*p* = 0.25). However, there was a significant difference in fetal heart rate between the groups during the second task period (*p* < 0.01) and the second recovery period (*p* = 0.04), to such an extent that fetuses whose mothers received the Stroop task first had higher HR during both periods (Figure 3).

Second, repeated measures ANOVA analyses were performed. Results from these analyses demonstrated that the main effect of the period (collapsed across both groups of fetuses, i.e., the baseline period versus the Stroop task period, the paced breathing task period, and the recovery periods) was not significant (*F* = 0.39, *p* = 0.81), suggesting that fetal HR did not change from the baseline period to either of the tasks or to the recovery periods. However, a significant task-by-group interaction indicated that fetal HR increased during the paced breathing task in fetuses whose mothers completed the Stroop task first (*p* < 0.01), but did not change over time in those whose mothers completed the paced breathing task first.

Furthermore, an ANOVA analysis compared the fetal HR values during each period to the preceding period. When the paced breathing task was presented to mothers first, there were no differences in fetal HR between periods. When the Stroop task was presented to mothers first, there was a significant increase in fetal HR only from the first recovery period to the second task period—the paced breathing task (*p* = 0.04) (Figure 3).

### 3.3. Maternal Responses to Tasks

Maternal HR. Maternal HR during the various periods are presented in Table 1 and visually depicted in Figure 4. Results from repeated measures ANOVA revealed a significant task-by-group interaction (*p* = 0.04). Specifically, women who received the Stroop task first increased their HR from baseline during the Stroop task, while the women who received the paced breathing task first did not increase their HR from baseline. No additional changes in maternal HR were observed across periods for either group. Maternal HR was not associated with fetal HR during any period (*p* values range from 0.08–0.74), except during the first recovery period (*r* = 0.24, *p* = 0.03).

Maternal Respiratory Rate. Maternal respiratory rates during the various periods are presented in Table 1 and Figure 5. Results from the repeated measures ANOVA suggest that women in both groups (Stroop task first and paced breathing task first) experienced sustained increases in respiratory rate from the baseline during the Stroop task (*p* < 0.01). Due to the nature of the paced breathing task (acute increases and decreases across the period), the average maternal respiratory rate during the paced breathing task did not increase from the baseline (*p* = 0.43). The task-by-group interaction term was not significant (*p* = 0.41), suggesting there were no group differences in maternal respiratory rate across the various periods. Maternal respiratory rate during each task was not correlated with mean fetal HR during that task (*p* values range from 0.15–0.84).

Maternal Stress Ratings. The average stress scores are provided in Table 1. Results from the repeated measures ANOVA suggest that ratings at baseline did not differ significantly between the groups of women (*p* = 0.31). A significant task-by-group interaction suggests that, for the women who received the Stroop task first, average stress levels during the two tasks did not differ. However, the women who received the paced breathing task first reported that the Stroop task was more stressful (*F* = 5.25, *p* = 0.02). Maternal stress ratings during each task were not correlated with fetal HR during that task.

### 3.4. Sensitivity Analyses of Group Differences in Fetal HR

In sensitivity analyses, we examined whether demographic and psychosocial variables accounted for, or moderated, the effect described above. Specifically, we examined whether maternal stress ratings of the tasks or fetal sex moderated the results. These variables were not independently or interactively related to fetal HR during the periods. Our findings were unchanged when each demographic variable presented in Table 1 was included in the model (each variable included in a separate model).

## 4. Discussion

The current study examined patterns of third-trimester fetal HR to determine if there is evidence of priming in utero. The counterbalanced laboratory study design allowed for the observation of different fetal HR patterns, each of which could support a unique hypothesis. First, there could be charactersistics of the paced breathing task, which the fetuses register and respond to with a change in HR. In the present study, this hypothesis seems less likely since fetal HR only increased during the paced breathing task when it followed maternal exposure to the Stroop task, suggesting that maternal exposure to the paced breathing task alone does not elicit significant changes in fetal HR. Second, there could be a cumulative effect of maternal engagement in laboratory tasks, which the fetuses recognize and respond to with an increase in HR. Since fetal HR did not reliably change during the second task, the cumulutive laboratory pertubation hypothesis is less likely to be accurate. Finally, the fetal HR pattern could support the identification of a priming effect, which occurs when exposure to one stimulus impacts the response to a subsequent, related stimulus [36]. Fetal HR increased during the paced breathing task only when the Stroop task was presented first. Furthermore, there was no group difference in fetal HR during the first recovery period. The group difference in fetal HR emerged later during the second task period when mothers completed another ANS-stimulating task similar to the first one. Taken together, these results suggest that fetal priming may exist; the prior experience of maternal engagement with a specific ANS-stimulating task elicited a fetal HR response when exposed again to a similar experience.

It is important to consider the changes in fetal HR in the context of the corresponding maternal responses in order to further understand the priming effect. In the present study, maternal heart rate increased from the baseline during the Stroop task, but only when the Stroop task was presented first. Maternal respiratory rate increased during the Stroop task, regardless of task order. In accordance with these results, the fetuses registered the maternal HR and respiratory rate reactivity to the Stroop task. This influenced their response during maternal engagement with a subsequent, related task involving acute changes in respiration, the paced breathing task. Conversely, when presented first, the paced breathing task did not elicit a sustained change in maternal HR or respiratory rate—though the maternal respiratory rate increased during the second Stroop task. Without a maternal HR or sustained respiratory rate response during the first paced breathing task, and thus without prior priming, fetuses did not show a HR response during the second related task, the Stroop task (despite a corresponding increase in maternal respiratory rate). Importantly, there was also no group difference in fetal HR during the first recovery period; the difference only emerged following the second exposure to an in utero pertubation via materal engagment in a laboratory task. This further demonstrates the specifity of the results and the action of priming. Finally, fetal HR changes did not track maternal HR or respiratory activity, which is consistent with previous work demonstrating maternal task-based physiology as a sensory stimulus for the fetus [24,37].

These findings are likely consistent with previous work suggesting that memory capacities emerge prior to birth. Habituation and classical conditioning, both forms of implicit memory, have been demonstrated in utero [10,11,13,14]. Our study is one of the few studies to suggest that fetuses exhibit a third form of implicit memory, priming. It is important to understand the development of in utero memory capacities, since previous work has suggested that these capacities may be associated with development in childhood. For example, the speed at which fetuses habituate is inversely related to their gestational age and postnatal outcomes [38,39,40,41]. Fetal priming may be of particular interest owing to its uniquely adaptive nature which elicits more efficient behavioral and cognitive responses to environmental stimuli. There is also evidence for more sophisticated fetal memory functions. Several studies have shown that flavors from the maternal diet are transferred to the fetus through amniotic fluid. After delivery, infants demonstrate recognition and even preference for these flavors [42,43]. Taken together, the findings across these studies suggest that in utero exposure to maternal stimuli, as well as fetal memory capacities, may influence long-term development, warranting further investigation.

The current study has several strengths. First, the study utilized data from an ethnically diverse group of women, more than half of whom identified as Latina, who remain understudied in developmental literature. This ethnically diverse group of women also improves the generalizability of our results to the broader population. Second, sensitivity analyses were performed and determined that maternal stress ratings of the tasks and fetal sex did not moderate the results. Furthermore, demographic variables outlined in Table 1 were included in the model and did not change the results. Finally, there are only several empirical investigations of fetal memory capacities involving larger sample sizes similar to the sample size in the present study [7,9,10,11,40].

There are limitations to the present study. Our data arose from a population of women living in a large urban area in the northeast; these results should be replicated with other populations. Further, the current study was not focused on identifying the specific priming stimulus or the mechanism that explains this priming effect. Future studies should investigate other maternal physiological processes (e.g., blood pressure changes) that may be responsible for our findings, as well as a finer resolution of maternal and fetal physiological changes. Such an investigation may provide insight into the perceptual processes involved in the early manifestations of priming.

In summary, the results of this current study suggest that there may be evidence for another form of implicit memory in utero, known as priming. Further investigations into the emergence of prenatal memory capabilities may provide insight into fetal cognitive capacities and the influence of the prenatal environment on later brain and behavioral development.

## Figures and Tables

**Figure 1 children-09-01670-f001:**
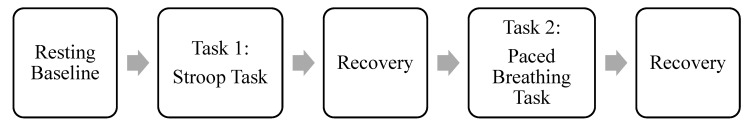
Overview of Study Procedures for Group 1.

**Figure 2 children-09-01670-f002:**
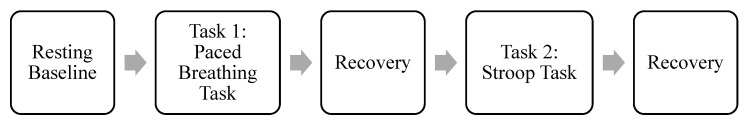
Overview of Study Procedures for Group 2.

**Figure 3 children-09-01670-f003:**
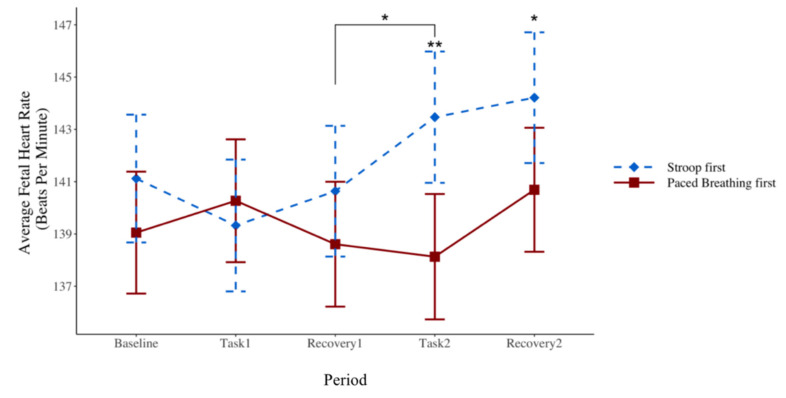
Fetal HR During Each Study Period By Group. * *p* < 0.05, ** *p* < 0.01.

**Figure 4 children-09-01670-f004:**
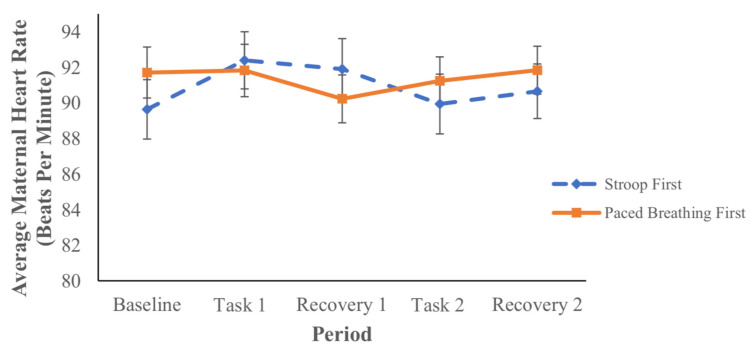
Maternal HR During Each Study Period By Group.

**Figure 5 children-09-01670-f005:**
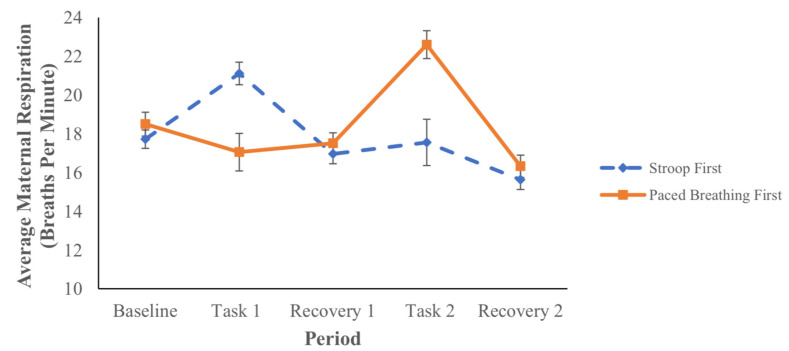
Maternal Respiratory Rate During Each Study Period By Group.

**Table 1 children-09-01670-t001:** Descriptive Statistics.

	*N* = 51	*N* = 54
	Stroop Task First (Group 1)	Paced Breathing Task First (Group 2)
	Mean (SD), *N*, or %	Mean (SD), *N*, or %
Ethnicity (% Latina)	62.75%	64.81%
Income		
$0–$15,000	6	3
$15,000–$25,000	9	10
$26,000–$50,000	8	14
$51,000–$100,000	11	16
$101,000–$250,000	15	7
Over $250,000	1	3
Age (years)	30.72 (6.43)	30.21 (5.72)
Education (years)	15.65 (3.46)	15.33 (3.02)
Child sex (% Female)	50%	51%
Pre-pregnancy BMI	25.50 (5.80)	25.45 (4.70)
GA at birth (weeks)	39.17 (1.42)	39.61 (1.12)
Birthweight (g)	3313.80 (434.81)	3468.56 (414.14)
GA at session (weeks)	34.39 (1.40)	34.59 (1.46)
Maternal Stress Ratings		
Baseline	1.73 (1.45)	2.19 (1.74)
Task 1	2.10 (1.64)	2.33 (1.39)
Task 2	2.40 (1.55)	3.76 (1.80)
Maternal Heart Rate		
Baseline	89.64 (11.80)	91.70 (10.33)
Task 1	92.39 (11.35)	91.23 (9.86)
Task 2	89.93 (11.88)	91.82 (10.41)
Maternal Respiration		
Baseline	17.72 (3.18)	18.50 (4.27)
Task 1	21.11 (3.35)	17.05 (6.65)
Task 2	17.55 (7.56)	22.60 (4.93)

Note: GA = Gestational Age. Two participants (one from each group) did not provide income information.

**Table 2 children-09-01670-t002:** Fetal Heart Rate Values Across All Experimental Periods.

	Baseline		Task 1		Recovery 1		Task 2		Recovery 2	
	Mean	*SD*	Mean	*SD*	Mean	*SD*	Mean	*SD*	Mean	*SD*
Full Sample	140.14	8.26	140.27	7.44	139.84	8.65	141.01	8.24	142.54	9.31
Paced Breathing First	139.13	6.12	140.65	7.00	138.61	7.55	138.17	7.01	140.75	9.36
Stroop First	141.21	9.99	139.82	7.99	141.13	9.58	143.97	8.47	144.49	8.95

## Data Availability

Data will be provided upon request.

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
