# Peer review of "Third Trimester Fetuses Demonstrate Priming, a Form of Implicit Memory, In Utero"

_children, 2022, doi:10.3390/children9111670_

Round 1

Reviewer 1 Report

Well written presentation of a very interesting topic. Good methodology description and discussion section;clearly noted study limitation. 

Reviewer 2 Report

Thank you very much for the opportunity to evaluate the submitted work. It presents a niche problem. The study design is well planned, conducted and presented. However, I have three comments, the introduction of which would help to dispel some doubts:

1. The authors do not refer to the basic dependencies of human physiology:

a) how the acceleration of respiration affects human HR under physiological conditions

b) how maternal acceleration of respiration and HR affect fetal HR

Outlining these relationships will help the reader understand that not only specific tasks can trigger these changes. They also react to specific daily tasks.

It is worth supplementing the introduction and discussions on these topics.

2. The authors write: "Second, the analysis of demographically accounted variables and additional variables that could moderate the results." However, I do not find measurement differences in the results due to these factors. So either delete this sentence or complete the results.

3. I would be careful in formulating unequivocal conclusions before verifying them with the comments in point 1 of the comments. I suggest using the phrases "probably" or " results show ".

In my opinion, the size of the research group, its characteristics and the lack of a control sample that does not perform the task or performs a neutral task do not allow for the formulation of an unambiguous opinion. Some of these comments are included in the restriction paragraph but suggest a softening of the conclusions stance.

Reviewer 3 Report

The topic was interesting, but the authors need to rectify some points in the manuscript.

1.         (Title) The title is too concise despite of small scale of the study. Could you add more information, such as study design and country name? In addition, this study does not directly deal with memory of fetuses. How about writing implicit memory or another word instead of memory in order to avoid misinterpretation?

2.         (Abstract) Could you write where this study took place?

3.         (Abstract) Please add statistical analysis methods used in the study.

4.         (Introduction) The association between fetal HR and memory is unobvious when I read the Introduction. It is not certain from a non-expert why changes in fetal HR leads to discussion about memory. Could you add more explanations or similar previous study in Introduction? I am not certain the word “memory” used in this study is broader compared with its general meaning.

5.         (Lines 87-88) Could you write more about the longitudinal study? Why the authors decided to use data of this longitudinal study?

6.         (Line 169) t-test and chi-squared tests are hypothesis testing, and “prior to hypothesis testing” is not correct.

7.         (Line 306) High percentage of latina women is a characteristics of the study. Could you write about generalizability of the results in Discussion?

8.         (Line 308-310) Is the sample size in this study large? I thought that the number of participants is relatively small as an epidemiological study. Could you cite similar fetal memory studies whose number of participants is small?
